# Netizens’ Food Safety Knowledge, Attitude, Behaviors, and Demand for Science Popularization by WeMedia

**DOI:** 10.3390/ijerph17030730

**Published:** 2020-01-23

**Authors:** Yong Zhao, Xinyang Yu, Yangxue Xiao, Zhengjie Cai, Xinmiao Luo, Fan Zhang

**Affiliations:** 1School of Public Health and Management/Research Center for Medicine and Social Development/Collaborative Innovation Center of Social Risks Governance in Health, Chongqing Medical University, Chongqing 400016, China; zhaoyong@cqmu.edu.cn (Y.Z.); aipiaodenvhai@163.com (X.Y.); 369xyx@sina.com (Y.X.); 2018111015@stu.cqmu.edu.cn (Z.C.); luoxinmiao215@163.com (X.L.); 2Research Center for Medicine and Social Development, Chongqing Medical University, Chongqing 400016, China; 3The Innovation Center for Social Risk Governance in Health, Chongqing Medical University, Chongqing 400016, China; 4Chongqing Key Laboratory of Child Nutrition and Health, Chongqing 400016, China

**Keywords:** netizen, food safety, knowledge, attitude, behaviors, WeMedia

## Abstract

This study aimed to investigate netizens’ food safety knowledge, attitudes and behavior, status, and their demand for science popularization by WeMedia. Firstly, participants were recruited by WeMedia, including WeChat, Microblog, and QQ. Then, a web-based survey was conducted using a self-designed questionnaire, which comprised 29 items about the knowledge, attitude, and practice (KAP) status of netizens’ food safety and the demand for science popularization by WeMedia. A correct answer on knowledge-related items was assigned a value of 1 point, and the high, medium, and low levels of knowledge had a total of 6–8, 3–5, and 0–2 points, respectively. A correct answer on attitude-related items was given a value of 1 to 5 points, and the high, medium, and low levels of attitude had a total of 16–20, 8–16, and l4–8 points, respectively. A correct answer on practice-related items was given a value of 5 to 1 point, and the high, medium, and low levels of attitude had a total of 22–30, 14–22, and 6–14 points, respectively. Results showed that the distribution of the different levels of the KAP scores were as follows: high (79.0%), medium (20.2%), low (0.8%); high (65.6%), medium (34.1%), low (0.3%); high (70.1%), medium (29.4%), low (0.5%). Approximately 86% of the subjects desired to obtain food safety knowledge from WeMedia. In conclusion, the netizens’ KAP in food safety are relatively optimistic. A large demand for science popularization on food safety knowledge by WeMedia is warranted. The WeMedia has a potentially important role in science popularization and health promotion related to food safety and health behaviors.

## 1. Introduction

Food is a human necessity. However, with the development of society, food safety has caused anxiety to human beings and, thus, has become a topic of interest [1].

According to a report released by the World Health Organization (WHO), approximately 600 million people worldwide suffer from eating contaminated food, resulting in 420,000 deaths per year. More than two-thirds of this demographic are young people (18–40 years old) [2]. Recently, in 2019, the FAO/WHO International Food Safety Authorities Network (INFOSAN) published a study protocol that focuses on food safety. In the publication, food safety was identified as a crucial issue. If a study on food safety can be conducted excellently, then it can provide comprehensive information about global food safety, including food safety-related events or experiences, because INFOSAN includes 188 countries. Such information will be beneficial for future practical and research work [3]. A report has stated that the burden of foodborne diseases is heaviest in the African and Southeast Asian regions [2], but the condition in China is not optimistic. In China, the residents’ life has been greatly improved in the past several years. Before, Chinese people cared more about whether residents had enough food to eat. Nowadays, food safety problems have become a new challenge for the Chinese government and the Chinese people [4,5]. Moreover, studies have focused on the changes regarding what China has considered about food safety over the past 40 years [6].

Usually, a series of procedures for food processing and transportation is present, during which the food may be contaminated and harmful food is produced. A harmful issue may occur once the food is consumed. A typical case was the baby milk powder incident that happened in 2008 involving contaminated milk powder, which caused the growth of kidney stones in several babies [7]. As reported, 2387 food safety issues, of which 99,487 people were poisoned and 380 died due to foodborne illness, have been reported in the past decade in China [8]. These numbers were the data from over five years ago. Currently, with the economic development, Chinese people regard food and health better than they did before. The focus regularly changes from enough food to safe food. What is food safety, and how do people obtain their information nowadays?

Among all the methods of obtaining information, WeBlog, WeChat, and other WeMedia tools have become popular in China due to the emergence of networks and the development of mobile phone applications [9,10,11,12]. The rapid dissemination and strong interactivity of these WeMedia tools have attracted an increasing number of people, and such tools have become the major channels through which people obtain information on food safety. However, the WeMedia company of China can hardly guarantee the validity of the contents they issue. Firstly, such observation is because some WeMedia companies only provide a platform for their consumers to issue information. The WeMedia company requires their consumers, who are also the information providers, to guarantee the validity of their information. The companies themselves focus on the operation of the platform rather than the quality of the issued information. To some extent, the companies neglect their responsibility for information validity. Secondly, some information providers are not professionals; this situation may lead to invalid food safety information. Incorrect information easily misleads and sometimes causes critical consequences, such as false news and public panic [13].

However, we can maximize the above characteristics of WeMedia to improve the netizens’ knowledge, attitude, and practice (KAP) whilst weakening its disadvantages. This study aims to obtain the current situation of netizens’ food safety KAP. On this basis, we identify their demand for scientific food safety knowledge through WeMedia, thus providing evidence for the popularization of food safety science through WeMedia.

## 2. Materials and Methods

### 2.1. Survey Subjects and Methods

The respondents for this research were netizens. We made an online questionnaire on the Wenjuanxing website and released it through the WeChat official account, personal WeChat moments, WeChat group, WeBlog, QQ space, and QQ group. We obtained 361 qualified participants.

With the development of technology, a new concept, namely, WeMedia, has emerged [14,15]. The descriptions of some WeMedia types are as follows: WeChat (in Chinese, it is pronounced as weixin, and the webpage users can access it through https://wx.qq.com/) is a communication service that can send text and voice messages. WeChat was developed by Tencent as QQ. Users access it mostly through the mobile phone, but they can also access it through the computer. The platforms include Android, iOS, and Windows. WeBlog (in Chinese, it is pronounced as weibo, and the webpage users can access it through http://t.qq.com/) is a communication tool where users can send messages of less than 140 words, which is less than the traditional blog. In this tool, users who follow you can receive the messages you post. Twitter is more popular outside of China. In China, the WeBlog issued by Tencent and Sina are the two most popular WeBlogs. For these WeMedia tools, users can choose to download the application on their mobile phone or computer or use its webpage to login.

### 2.2. Survey Questionnaire

Designed by our team members, the online survey questionnaire comprised the following three parts:(1)Demographic and economic information: six items, including gender, age, ethnicity, registered residence, education level, and household per capita monthly income. Please see Table 1 for details.(2)Food safety KAP: 18 items.

The eight knowledge items are shown in Table 2. The following is an example as ‘The raw food and cooked food should be stored and processed separately’. Those who answered correctly would score 1 point for each item. The subtotal score was divided into three levels: high (6–8 points), medium (3–5 points), and low (0–2 points).

The four attitude items are shown in Table 2. The following is an example as ‘You are worried about vegetable pesticide residues’. The options range from ‘strongly agree’ (5 points) to ‘strongly disagree’ (1 point). The subtotal score was divided into three levels: high (16–20 points), medium (8–16 points), and low (4–8 points).

The six behavior items are shown in Table 2. The following is an example as ‘View food labels (such as date of manufacture, shelf life, ingredients, nutrients, etc.) when shopping online’. The options ranged from ‘always’ (5 points) to ‘never’ (1 point). The subtotal score was divided into three levels: high (22–30 points), medium (14–22 points), and low (6–14 points).

(3)Status of WeMedia use and demand for science popularization: five items, including ‘the most commonly used WeMedia’, ‘the willingness to obtain food safety knowledge through WeMedia’ and ‘the way and method of obtaining it’, and ‘whether to pay attention to food safety information through WeMedia’.

In addition to the three parts above, the survey tool also included an informed consent form.

### 2.3. Quality Control

The questionnaire was repeatedly discussed and revised by the experts in the team members and in the field. A pilot survey was conducted to obtain pieces of advice for the revision of specific items and the understanding and support of the respondents. The respondents volunteered to participate, which may imply that they would provide a true answer.

### 2.4. Statistical Methods

Questionnaire data were exported from the survey website as an Excel file. Statistical software SPSS 22.0 was used for data analysis. Measurement data were expressed as mean ± standard deviation, and categorical data were expressed in terms of frequency and percentage. Based on the results of the normality test, the food safety KAP scores did not agree with the normal distribution (sig. = 0.000 < 0.05); hence, the rank sum test (single factor analysis) was used. Generalized linear models (multivariate analysis) were used to analyze the risk factors of KAP. A two-sided test *p* < 0.05 was considered statistically significant.

## 3. Results

### 3.1. General Demographic and Economic Characteristics

The distribution of each demographic and economic characteristic of the 361 netizens is shown in Table 1.

### 3.2. Status of Food Safety Knowledge, Attitudes, and Behavior

The distribution of the total score of the knowledge on food safety was high (79.0%), medium (20.2%), and low (0.8%). The distribution of the total score of the attitude on food safety was high (65.6%), medium (34.1%), and low (0.3%). The distribution of the total score of the behavior on food safety was high (70.1%), medium (29.4%), and low (0.5%). The average score of food safety knowledge of the respondents was 6.22 ± 1.14, the average score of food safety attitude was 17.48 ± 2.38, and the average score of food safety behavior was 24.19 ± 3.74.

In the behavior section, respondents always or frequently check the sensory condition of the food when purchasing food (58.7%, 32.7%), check the food label when shopping for food online (49.9%, 31.6%), preserve the food according to storage conditions (34.6%, 35.7%), wash hands before washing the food (43.2%, 36.6%), and eat fruits after washing and peeling (58.4%, 26.9%). However, in the event of food safety problems, merchants were just sometimes or occasionally required to deal with or complain to relevant departments (36.8%, 21.3%).

### 3.3. Risk factors of Food Safety Knowledge, Attitudes, and Behavior

Univariate analysis (rank sum test) showed that significant differences in food safety KAP scores amongst different ages, household registration, and household per capita monthly income were present (*p* < 0.05). Moreover, significant differences in food safety knowledge scores between different levels of education were observed (*p* < 0.05). Please see Table 3 for more detail.

Multivariate analysis (generalized linear model) showed that the risk factor for knowledge score was the educational level (junior high school and below) (Table 4). In addition, the risk for the behavior was the education level (college or vocational college) (*p* < 0.05).

### 3.4. Status of Use of WeMedia and Demand for Science Popularization

The most used WeMedia for the respondents were WeChat (97.51%) and QQ (80.06%). In addition, 51.25% of the respondents really hoped to obtain knowledge from WeMedia. Amongst all WeMedia applications, 60.56% preferred the WeChat public number (60.56%). For the forms of the WeMedia, 91.55% of the respondents preferred the combination of graphic and text, followed by video (59.15%) and comics (42.54%). Please see Table 5 for further details.

## 4. Discussion

In the past years, several online platforms for takeaway food have been developed, thereby causing people to regard food safety [16,17]. The government has already issued some national standards for food safety, and some studies have been conducted to obtain the knowledge status of related government personnel [18,19]. All these reports show that food safety has become an important topic in China.

According to the 41st Statistical Report on China’s Internet Development [20] released by the China Internet Network Information Centre (CNNIC), as of December 2017, the number of Internet users in China reached 772 million, amongst which the rural netizens accounted for 27.0%, and urban netizens accounted for 73.0%. Mobile internet users accounted for 97.5%, and the age structure of netizens was dominated by 10–39 years old, which accounts for 73.0% of the total. The netizens within the 20–29 age group accounted for the highest proportion, reaching 30.0%. The netizens with a monthly income at the medium and high levels accounted for the highest proportion, and the monthly incomes of 2001–3000 and 3001–5000 accounted for 16.6% and 22.4%, respectively. The proportion of netizens with junior high school, high school/secondary school/technical school qualifications was 37.9% and 25.4%, respectively. In general, based on the results of this study, the population in this study is representative of the netizens. In this study, rural netizens accounted for approximately 38%, which is higher than the reported number by the CNNIC. This observation may be due to the study being conducted later than the time of the CNNIC report. As time passed, more rural people have started to use the Internet. In this study, almost 70% of the survey population were aged under 40 years old, which is mostly consistent with the report by the CNNIC. In this study, 72.9% of the subjects had an educational level of undergraduate and above, which was higher than the report by the CNNIC. This observation is probably because the online questionnaire was mainly promoted by graduate students and undergraduates in their own communication circles. Most people in their circles were those working or studying in colleges and universities, thus resulting in a high overall educational level of the survey population. In addition, for the monthly outcome, the classification methods of this study and those of the CNNIC were different. Hence, we did not compare it.

The food safety knowledge, attitudes, and behavior of the netizens in this study were good. This is consistent with some studies conducted in other counties. For example, one study was conducted among university students in Iran. It showed that over half of the students had high knowledge and attitude. However, in some other reports, the result is different [21]. Firstly, this observation may be related to the high overall educational level of the netizens in the survey. Secondly, the items in the questionnaire in this study may not be totally the same as other reports or papers. In addition, amongst the food safety behavior asked in this study, some interventions can be designed and implemented on the basis of the answer to the next question. Amongst all the food safety behavior, this behavior needed the most improvement. When netizens encounter food safety problems, they are less aware of the need for merchants to deal with or complain to relevant departments. A study in China [22] has shown that people are generally low in food safety awareness, and only 36.8% said they were willing to complain to the relevant departments after buying adulterated food. One possible reason is that some people are afraid of trouble. Another reason may be that the complaint procedure is too cumbersome or even unknown, making people less willing to maintain their legitimate rights and interests. Moreover, people are not aware that they can report their experience and ask for justice from food safety authorities. Some of these reasons are also supported in Luo’s study [22].

The correlation analysis in this study showed that the higher the food safety knowledge scores, the better attitude and behavior; this relationship conforms to the health education model theory. The change of knowledge and attitude is the basis of behavior change [23]. In general, the deeper the knowledge, the stronger the tendency to implement. However, through a literature review, we found that the relationship between knowledge and behavior is not always a positive correlation.

On the basis of the univariate analysis in this study, no significant difference amongst males and females for the knowledge, attitude, and behavior was observed. This observation is different from the results of some studies. Most studies have reported that the KAP of females were better than males [23,24,25]. A study mentioned that females were more thoughtful, paid more attention to their own health and studying knowledge [23]. Another study mentioned that compared with males, females usually perform more work in family diets, and residents with higher education levels can easily understand food safety related knowledge [26]. For ages, the food safety knowledge level of those older than 40 has been better than those younger than 40 probably because, as time passed, people began to be aware of the importance of health and food safety. Those with urban household registration have better food safety knowledge, which may be due to numerous opportunities for health education [26]. However, residents with rural household registration are important for food safety. Possibly, most of the family members of these residents are farmers, or they changed from farmers to factory workers. A previous study mentioned that farmers are important for food safety because they may join the food production procedures. Moreover, farmers are food consumers and may be exposed to unsafe food [27]. The higher the household monthly income per capita, the better food safety knowledge because a good economic situation may provide favorable education opportunities; thus, further attention is paid to food safety [22]. In addition, people with different education levels have different food safety knowledge, but no statistical difference in food safety attitude and behavior scores was observed, indicating that a regular food safety training is necessary amongst community residents [26]. Furthermore, the multivariate analysis results showed that the higher the educational level is, the higher the food safety knowledge and behavior scores of the respondents are. This observation is consistent with the findings of some studies in China [25,28,29]. Briefly, the factors above can be considered when health education strategies are designed.

The most used media for respondents were WeChat (97.51%) and QQ (80.06%). According to the 41st Statistical Report on China’s Internet Development Status released by the China Internet Network Information Centre (CNNIC), as of December 2017, the usage rate of WeChat friends circle, QQ space, and WeBlog users were 87.3% and 64.4%, 40.9%, respectively [20]. Almost 90% of the respondents really hoped or hoped to obtain food safety knowledge through the media, and over half of the netizens preferred the WeChat public account. This observation may be because the netizens in China use WeChat frequently, and WeChat public is convenient to obtain information. As such, food safety knowledge can be issued by the WeChat public account, combined with other commonly used self-media. Also, over 90% of the respondents prefer obtaining food safety related knowledge through graphic and textual (91.55%), followed by video (59.15%) and comics (42.54%). This observation may be because the information search behavior had the characteristics of horizontality, jumping, inspection, and browsing [30]. The combination of graphic and text forms is in line with the habit of searching and browsing information. According to the survey, netizens expressed their willingness to take the initiative to understand food safety knowledge and hoped to obtain relevant knowledge through multiple channels, which build a good foundation for the transition from attitude change to behavior change [28].

The research population included netizens. Hence, we used the online questionnaire in this study. Although no occupational question was present in our survey, we could infer from the results that this may cause bias in the validity of our results, for example, we obtained a high knowledge score. The reason may be that the online questionnaire was mainly distributed by graduate students and undergraduates of the Department of Food Hygiene and Nutrition of Chongqing Medical University in their own communication circles (mainly colleges and universities) and encouraged netizens to participate, resulting in a high overall educational level of the survey population. Future studies should involve additional occupations.

## 5. Conclusions

This study showed that netizens’ food safety knowledge, attitudes, and behavior are satisfying. Netizens have a large demand for food safety knowledge from the WeMedia. When government officials conduct food safety science popularization, WeMedia may be a potential or even important way. Similarly, in other health or social fields, the use of WeMedia may also have potential importance. Multiple WeMedia methods could be used together without limitation to only one method. Moreover, promoting the improvement of education level is always an important work for the government, which may be beneficial for future improvements of food safety or other social problems.

## Figures and Tables

**Table 1 ijerph-17-00730-t001:** Distribution of demographic characteristics (n = 361).

Variable		Case Number (*n* = 361)	Percentage (%)
Gender	Male	104	28.8
	Female	257	71.2
Age	<40	245	67.9
	≥40	116	32.1
Nationality	Han	312	86.4
	Others	49	13.6
Household registration	Township	223	61.8
	Rural	138	38.2
Educational level	Junior high school and below	21	5.8
	Highschool/secondary school	25	6.9
	College/vocational college	52	14.4
	Bachelor degree or above	263	72.9
Household per capita monthly income	≤4000	122	33.8
	4000–8000	121	33.5
	≥8000	118	32.7

**Table 2 ijerph-17-00730-t002:** Description of food safety behavior (*n* = 361).

Knowledge Item							True		False/No Idea	
							*n*	%	*n*	%
Food can be kept in the refrigerator for a long time.							331	91.7	30	8.3
Raw food and cooked food should be stored and processed separately.							357	98.9	4	1.1
Cooked food or leftovers should be thoroughly heated before eating.							326	90.3	35	9.7
Not fully cooked green beans can’t be eaten.							342	94.7	19	5.3
“Shelf life” means the period in which prepackaging food maintains quality.							278	77.0	83	23.0
Spices and flavors can be used in milk.							61	16.9	300	83.1
Food borne diseases are infectious or toxic.							272	75.3	89	24.7
The online shopping food has been included in the scope of supervision of the new Food Safety Law of the People’s Republic of China.							277	76.7	84	23.3
**Attitude Item**	**Always**		**Frequently**		**Sometimes**		**Occasionally**		**Never**	
	***n***	**%**	***n***	**%**	***n***	**%**	***n***	**%**	***n***	**%**
Food safety incidents in China ^1^	161	44.6	99	27.4	96	26.6	3	0.8	2	0.6
Pesticide residues in vegetables ^2^	198	54.8	109	30.2	52	14.4	1	0.3	1	0.3
Veterinary residues of meat ^3^	204	56.5	101	28.0	52	14.4	3	0.8	1	0.3
Inappropriate food safety behaviors change ^4^	216	59.8	126	34.9	18	5.0	1	0.3	0	0
**Behaviors Item**	**Always**		**Frequently**		**Sometimes**		**Occasionally**		**Never**	
	***n***	**%**	***n***	**%**	***n***	**%**	***n***	**%**	***n***	**%**
Check the sensory ^5^	212	58.7	118	32.7	29	8.0	2	0.6	0	0
View food labels ^6^	180	49.9	114	31.6	48	13.3	15	4.2	4	1.1
Require merchants ^7^	56	15.5	42	11.6	133	36.8	77	21.3	53	14.7
Preserve food ^8^	125	34.6	129	35.7	83	23.0	21	5.8	3	0.8
Wash hands ^9^	156	43.2	132	36.6	53	14.7	20	5.5	0	0
Eating fruits ^10^	211	58.4	97	26.9	35	9.7	15	4.2	3	0.8

^1^: You are concerned about food safety incidents in recent years in China; ^2^: You are concerned about pesticide residues in vegetables; ^3^: You are concerned about the veterinary residues of meat; ^4^: You are willing to change inappropriate food safety behaviors; ^5^: Check the sensory condition of food when purchasing food; ^6^: View food labels when shopping online; ^7^: Require merchants to deal with or complain to relevant departments when food safety problems are encountered; ^8^: Preserve food according to storage conditions; ^9^: Wash hands before taking food; ^10^: Eating melon and fruits after washed or peeled thoroughly.

**Table 3 ijerph-17-00730-t003:** Univariate analysis of knowledge, attitude, and practice (KAP) risk factors related to food safety (*n* = 361).

	Knowledge Scores			Attitudes Scores			Behaviors Scores		
	Mean ± SD	Z/χ^2^	*p*	Mean ± SD	Z/χ^2^	*p*	Mean ± SD	Z/χ^2^	*p*
Male	6.06 ± 1.17	−1.77	0.077	17.63 ± 2.49	−1.12	0.260	23.93 ± 4.05	−0.57	0.570
Female	6.28 ± 1.13			17.42 ± 2.33			24.30 ± 3.61		
<40 ^1^	6.10 ± 1.18	−2.85	0.004 *	17.16 ± 2.41	−3.90	0.000 *	23.70 ± 3.83	−3.65	0.000 *
≥40 ^1^	6.47 ± 1.01			18.16 ± 2.17			25.24 ± 3.31		
Han ^2^	6.27 ± 1.08	−1.51	0.130	17.54 ± 2.39	−1.38	0.169	24.32 ± 3.74	−1.79	0.074
Others ^2^	5.88 ± 1.42			17.10 ± 2.31			23.37 ± 3.66		
Township ^3^	6.39 ± 1.02	−3.46	0.001 *	17.74 ± 2.37	−2.95	0.003 *	24.89 ± 3.45	−4.45	0.000 *
Rural ^3^	5.93 ± 1.27			17.06 ± 2.33			23.07 ± 3.91		
Junior ^4^^a^	5.52 ± 1.37	9.21	0.027 *	16.76 ± 2.43	7.27	0.064	24.14 ± 4.04	7.13	0.068
High ^4^^b^	6.00 ± 1.47			17.56 ± 1.90			23.44 ± 3.60		
College ^4^^c^	6.17 ± 1.04			18.27 ± 1.87			25.29 ± 3.77		
Bachelor ^4^^d^	6.30 ± 1.09			17.37 ± 2.48			24.05 ± 3.70		
≤4000 ^5^	5.98 ± 1.15	14.37	0.001 *	17.11 ± 2.36	7.30	0.026 *	23.42 ± 3.79	19.20	0.000 *
4000−8000 ^5^	6.17 ± 1.25			17.45 ± 2.43			23.90 ± 3.68		
≥8000 ^5^	6.51 ± 0.94			17.89 ± 2.30			25.30 ± 3.50		

^1^: Age; ^2^: nationality; ^3^: household registration; ^4a^: education level of junior high school and below; ^4b^: education level of high school/secondary school; ^4c^: education level of college/vocational college; ^4d^: education level of bachelor degree or above; ^5^: household per capita monthly income; * *p* < 0.05.

**Table 4 ijerph-17-00730-t004:** Multivariate analysis of KAP risk factors related to food safety (*n* = 361).

Variable	Knowledge Scores		Attitudes Scores		Behaviors Scores	
	β (s.e.)	*p*	β (s.e.)	*p*	β (s.e.)	*p*
Male vs. Female	0.24(0.13)	0.059	−0.48(0.28)	0.080	0.01(0.43)	0.974
<40 vs. ≥40 ^1^	0.26(0.14)	0.071	0.04(0.31)	0.900	0.50(0.49)	0.304
Han vs. others ^2^	−0.10(0.18)	0.580	−0.32(0.40)	0.427	−0.90(0.62)	0.148
Rural vs. Township ^3^	−0.19(0.15)	0.184	−0.10(0.31)	0.740	−0.06(0.49)	0.911
Junior high school and below ^4^	−0.60(0.26)	0.018 *	0.27(0.55)	0.629	−0.07(0.86)	0.935
High school/secondary school ^4^	−0.30(0.23)	0.205	−0.42(0.51)	0.413	−0.74(0.79)	0.353
College/vocational college ^4^	−0.23(0.17)	0.190	−0.01(0.37)	0.990	−1.30(0.58)	0.026 *
≤4000 vs. ≥8000 ^5^	−0.25(0.16)	0.125	−0.10(0.35)	0.775	0.05(0.54)	0.920
4000−8000 vs. ≥8000 ^5^	−0.23(0.15)	0.107	−0.02(0.31)	0.941	−0.30(0.49)	0.546

^1^: Age with reference group as ≥40; ^2^: nationality with reference group as others; ^3^: household registration with reference group as township; ^4^: education level with reference group as bachelor degree or above; ^5^: household per capita monthly income with reference group as ≥8000; * *p* < 0.05.

**Table 5 ijerph-17-00730-t005:** Status of WeMedia use and demand for science popularization (n = 361).

Item		Total Population, n(%)
What are the most commonly used WeMedia in the last month? (multiple choice)	WeChat	352(97.51%)
	Sina WeBlog Tencent WeBlog	132(36.57%) 35(9.7%)
	QQ	289(80.06%)
	Blog	7(1.94%)
	Renren net	5(1.39%)
	Zhihu	31(8.59%)
	Tieba	22(6.09%)
	Others	13(3.6%)
Do you want to get food safety knowledge from the WeMedia?	Really hope	185(51.25%)
	Hope General	126(34.9%) 44(12.19%)
	Do not wish	5(1.39%)
	Strongly unwilling	1(0.28%)
Which kind of WeMedia do you most want to receive food safety knowledge in which we can achieve WeMedia intervention?	WeChat public number WeChat circle WeChat group QQ group QQ space	215(60.56%) 61(17.18%) 20(5.63%) 11(3.1%) 6(1.69%)
	Sina WeBlog	42(11.83%)
What form do you want to get relevant knowledge? (multiple choice)	Graphic combination	325(91.55%)
	Recording	57(16.06%)
	Video	210(59.15%)
	Comic	151(42.54%)
	Others	5(1.41%)

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
