# Peer review of "Netizens’ Food Safety Knowledge, Attitude, Behaviors, and Demand for Science Popularization by WeMedia"

_ijerph, 2020, doi:10.3390/ijerph17030730_

Round 1

Reviewer 1 Report

Overall, the manuscript is interesting and will add to the body of knowledge. The results may lead to changes in the delivery of food safety information in China and other countries. 

Please make the following changes to the manuscript.

The abstract as written is quite confusing. It is highly recommended that the abstract be re-written to be more clear and concise, as the context in the abstract is not descriptive enough to support the data. It may be best to leave out the scores and stick to the percentages. Line 40: "food safety incidents" is unclear. Please define what an "incident" is.   There are no descriptions/definitions of We-Chat, We-Media Microblog, etc. Please add descriptions of what they are and how they are used. Additionally, the capitalization/hyphen usage of these is inconsistent  in the manuscript.  Line 76-89: There needs to be better descriptions of the questions asked for each category (e.g. behavior, attitude, knowledge). It is difficult to determine the relevancy of the scores when the questions asked of the participants is unknown. Since there are only 18 questions, it would be good to include a graph or table describing each question and the overall average results. This was partially completed in table 3, but it would be good to expand it for all KAP categories. Table 1: The footnote is very confusing and I am very unclear what is trying to be stated. This needs to be re-worded. Line 120: "...preserve the food according to storage conditions" It is very unclear what this means. Is it meant that it was stored properly, or stored per manufacturer's directions? Please re-word. Line 139-140: States that the risk behavior was education level as college or vocational; however, if the score is always (5) to never (1) then wouldn't a high score indicate better compliance (e.g. always or usually)? There is an inconsistency between what is described int he materials and methods and the results. Please clarify or correct so the data is understood properly. Line 146-150: Please provide descriptions of We-Media applications. Those outside of the country are completely unaware of what these are and thus cannot evaluate the data or context properly.  Discussion: It would be best to also compare this data to data obtained in other countries. As noted, extensive editing of English language and style are required. General comment: Do not start sentences with the word "and" as it is improper. 

Author Response

Dear Reviewer,

We thank you for their valuable comments, which have helped us a lot when we revised and improved the manuscript. We have listed all the comments below and present a point-by-point response to the comments.

Besides, we refer to a company to help us to edit the English language by native English speakers. That took four days, which make our revision time longer and send you the feedback until the end of December. We really hope that our paper is pleasant to read now.

Please see the attachment for revision details. Thank you very much.

Fan ZHANG

Reviewer 2 Report

The paper contains information about netizens' food safety knowledge, attitudes and behaviors status and their demand for science popularization by We-Media.  

A regional study like this could provide a useful additional case study to extend the literature on food safety knowledge. However, there are a number of weaknesses in the current article and it is not suitable for publication in its current form. The paper suffers from multiple problems that prevent me from recommending its publication.

The most confusing is the lack of proper description of the methods and conclusion. There are too much theoretical description in the discussion section. In addition, the conclusions are the statements.  

I would like to know why it is so important to reveal the food safety knowledge, attitude and practice of Netizens in China. Are foodborne diseases the reason only?

The statistical analysis are very simple and I see more potential scientific value in the collected data.

The current description of the results must be rewritten because the results are not presented sufficient in the present form. Please move the demographic and economic characteristics to the Material and methods section. The manuscript provides a snapshot of food safety KAP. I recommend to present the results in a broader perspective. If you did it your paper would be more valuable for readers.

Please present the  questionnaire in the section 2.2.

I did not recognize any possibility of using the research in practice. Please try to consider it.

Please ensure the positive impact of your findings and experience on Netizens all over the world.

There are a lot of typing errors. E.g. L. 38, L. 40, L. 71, L.104, L. 145 and many others. Please correct them.

In my opinion the discussion section is too long. A lot of given information are redundant. Please move some of them to the introduction section.

There is more explanation needed in term of such good results concerning food safety KAP. In the L.40 the authors stated that the condition in China concerning the foodborne diseases is not optimistic. Please provides a common sense explanation of distributing the questionnaire among the graduate and undergraduate students of the Department of Food Hygiene and Nutrition of Chongqing Medical University. The results are predictable in comparison to other people.

Author Response

(The authors gave the same response as above.)

Round 2

Reviewer 2 Report

Dear Authors

Many thanks for the enhancement your paper and your effort. Unfortunately, you did not make all changes. However, the paper was improved. One issue is very important for me and still remain not improved. I did not recognize any changes in the conclusion section. The conclusion are the statements still. Please present the conclusions in a broader perspective.

Kind regards,

Reviewer

Author Response

Response to Reviewer 2 Comments

Point 1: Many thanks for the enhancement your paper and your effort. Unfortunately, you did not make all changes. However, the paper was improved. One issue is very important for me and still remain not improved. I did not recognize any changes in the conclusion section. The conclusion are the statements still. Please present the conclusions in a broader perspective.

Response 1: On behalf of my co-authors, we thank you very much for giving us an opportunity to revise our manuscript further to make it better. We thank you for this comment. We tried our best to give the conclusion in a broader perspective. After discussion among the authors, we delete some statements which can be shown clearly from the results and more detailed for the conclusion section. Then we add more conclusion description with broader perspective. The detail of the revision is present in the conclusion section of the full-text as well as the conclusion section of the abstract. We believe the conclusion section are much better now. Your insist on this comment make us know better how to write a better conclusion and how to show the significance of our study. Thank you very much for this comment.